# A Machine Learning Algorithm Predicting Infant Psychomotor Developmental Delay Using Medical and Social Determinants

David Waynforth

School of Medicine, Faculty of Health Sciences, Bond University, Gold Coast, QLD 4226, Australia;
dwaynfor@bond.edu.au

**Abstract:** Psychomotor developmental delay in infants includes failure to acquire abilities such as sitting, walking, grasping objects and communication at the ages when most infants have acquired these abilities. Known risk factors include a large number of aspects of family environment, socioeconomic position, problems in pregnancy and birth and maternal health. It is clinically useful to be able to screen for developmental delay so that healthcare interventions can be considered. The present research used machine learning (random forest) to create an algorithm predicting psychomotor delay in 9-month-old infants using information ascertainable at birth and in early infancy. The dataset was the UK longitudinal Millennium Cohort study. In total, 53 predictors measuring socioeconomic indicators, paternal, family and social support for the mother, beliefs about good parenting, maternal health, pregnancy and birth were included in the initial algorithm. Feature reduction showed that of the 53 variables, birthweight, gestational age at birth, pre-pregnancy BMI, family income and parents' ages had the highest feature importance scores and could alone correctly predict developmental delay with over 99% sensitivity and 100% specificity. No features measuring aspects of early infant care or environment meaningfully added to algorithm performance. The relationships between delay and some of the predictors, particularly income, were nonlinear and complex. The results suggest that the risk of psychomotor developmental delay can be identified in early infancy using machine learning, and that the best predictors are factors present prior to and at birth.

**Keywords:** developmental milestones; artificial intelligence; classification algorithms; infant growth

## 1. Introduction

Children's progress in achieving developmental milestones in infancy and childhood is dependent on a large number of factors. These include growth in utero, size at birth, maternal health, socioeconomic position, genetically inherited developmental patterns, and many family and social factors [1–17]. This makes predicting developmental delay in advance so that steps can be taken to avoid it difficult, as there are so many potentially important causes, and the relative importance of each is not clear. For an increasing number of health conditions with complex aetiologies, artificial intelligence (AI) has been successfully applied to identify when an individual is at high risk for a future adverse health outcome, e.g., [18,19]. In the discipline of developmental psychology, the machine learning approach of random forests (RF) has been applied to predict future psychiatric conditions [20] and to predict infant growth using inflammatory markers [21]. The present study applied RF to predict psychomotor developmental delay in 9-month-old infants using data on a wide array of factors in pregnancy, birth and early infancy. The intent was to achieve higher sensitivity and specificity than has been achieved in prior studies approaching similar problems using regression methods, which rarely have greater than 80% sensitivity [15].

### 1.1. Predictors of Developmental Delay

Developmental delay has been found to be statistically associated with an array of factors both temporally (prior to pregnancy, during the perinatal period and later in infancy)

and from both maternal and foetal causes. Preterm birth and low birthweight have been demonstrated to increase the risk of delay by double or greater [11–14]. Maternal factors including health, gravidity, young motherhood, use of assisted reproductive technologies and bodyweight have also been found to predict delay [8,12–16]. Other factors with large statistical effects on the likelihood of developmental delay include maternal education and socioeconomic position [12,13,15,17]. Some of the reported effects are large; for example, Ozkan et al. found more than a tenfold increased risk of developmental delay in infants of mothers with the lowest level of completed education [13]. The neonatal period and early childcare may also be important determinants of delay, including social support provided, paternal involvement, breastfeeding and other infant care behaviour [1–7].

### 1.2. Data Analytic Approach

Several statistical techniques are potentially appropriate for classification problems including predicting developmental delay using a large number of predictors. Most commonly, regression-based approaches have been used. Van Dokkum et al. [15] used logistic regression to predict developmental delay at age four, producing an algorithm with 73% sensitivity and 80% specificity. Another promising linear modelling approach when there is a large number of predictor variables is principal component analysis (PCA). However, both statistical techniques assume linear relationships between values of the predictor variable and the outcome: PCA is based on linear transformation using orthogonal matrices, and logistic regression assumes that the log-odds of the relationship between each predictor and the outcome is linear. There is no reason to believe that predictors have linear associations with developmental delay: for example, birthweight has negative associations with developmental delay at both low and very high levels [15,22], and socioeconomic position may not be important for health outcomes above a threshold level [23]. For the present research, random forest (RF), which is an ensemble decision-tree classifier, was chosen. RF can handle large numbers of predictors (features) simultaneously and does not assume linear or monotonous relationships between predictors and an outcome [24,25]. While most AI approaches have a barrier to entry in that they cannot be implemented in statistical programs commonly used by researchers, RF is straightforward to implement in Stata, as well as in some open-source statistical software such as BlueSky Statistics. Here, RF was applied to an existing national dataset on the birth and lives of a sample of infants born in the UK, the UK Millennium cohort.

## 2. Methods

### 2.1. Population and Sample

The UK Millennium cohort sample (henceforth MCS) consists of infants born in the United Kingdom from September 2000 to August 2001, identified using Universal Child Benefit records and NHS Health Visitors [26]. In the British healthcare system, Health Visitors are usually registered nurses who provide ante- and post-natal care and advice in the home. The sample was not a random sample: ethnic minority and low socio-economic groups were oversampled to compensate for loss to follow-up of these segments of the population that occurred in Britain's earlier longitudinal cohort studies. Here, data were analysed using the first survey of the cohort, which took place when the infants were around 9 months old. The maximum possible sample size for analysis using this cohort is 18,467. A cohort profile is available providing far more detail about the sample and sampling methods [27].

### 2.2. Outcome Variable

Developmental delay is typically identified in clinical settings using parental questionnaires. The 9-month MCS interview with parents or the main care giver included questions about infant psychomotor development which are very similar in content and format to the Ages and Stages 12-month questionnaire [28]. The aim in creating the dependent variable was to capture infant development across a number of cognitive and motor skill domains. Second, variation in reaching developmental milestones has the most practical or clinical significance if a statistical model is created to predict substantial delay versus the

range of normal development. With these aims in mind, a dependent variable was created using parental or main caregiver reports of achievement of developmental milestones. The interview contained 12 questions on cognitive and motor skills development. Responses to the 12 questions were on three-point scales, coded as "1" for the infant frequently demonstrates the developmental milestone, "2" for sometimes, and "3" for the infant has not yet demonstrated the milestone. The 12 items were the following: sits up; smiles; stands up holding on; puts hands together; grabs objects; holds small objects; passes a toy; walks a few steps; gives toy; waves bye-bye; extends arms; nods for yes. The responses were summed into a single score, followed by statistical correction for age in days of the infant. The resulting standardised residuals representing age-corrected developmental delay were then split into a binary variable using two standard deviations as the cut-point.

### 2.3. Predictor Variables (Features)

The first MCS survey was broad in scope, covering aspects of pregnancy, labour, birth and children's and their parents' social, work and economic situations. Many of the variables included in the MCS have been demonstrated to be or could plausibly be associated with child development. Covariates were selected by reading through the MCS variable list and selecting all that appeared appropriate for analysis. The variable selection process is illustrated in Figure 1. Some additive combining of variables was performed where two or more variables were repeated information about a single concept. For example, paternal involvement in infant care was represented in the original data as questions about each individual act of care, such as nappy changing, getting up in the night, etc. These were additively combined to create a single variable. Of note, a decision was made to combine medical problems in pregnancy into a single variable. In descending order of their prevalence in the dataset, the most common were as follows: bleeding in pregnancy, eclampsia, hyperemesis, urinary tract infections, anaemia, and non-trivial infections. These were combined because, conceptually, they should all affect foetal nutrition, and because in initial testing of algorithms they performed very poorly as predictors of developmental delay when included separately. In total, 53 variables were included. For ease of reading, variables were classified into groupings based on the concept that each represented: family and social support; socioeconomic indicators; infant characteristics; beliefs about parenting; medical circumstances in pregnancy and birth; maternal factors; and paternal and family factors. Supplementary material Table S1 includes details of variable coding, the MCS names and any changes made to the original MCS variables.

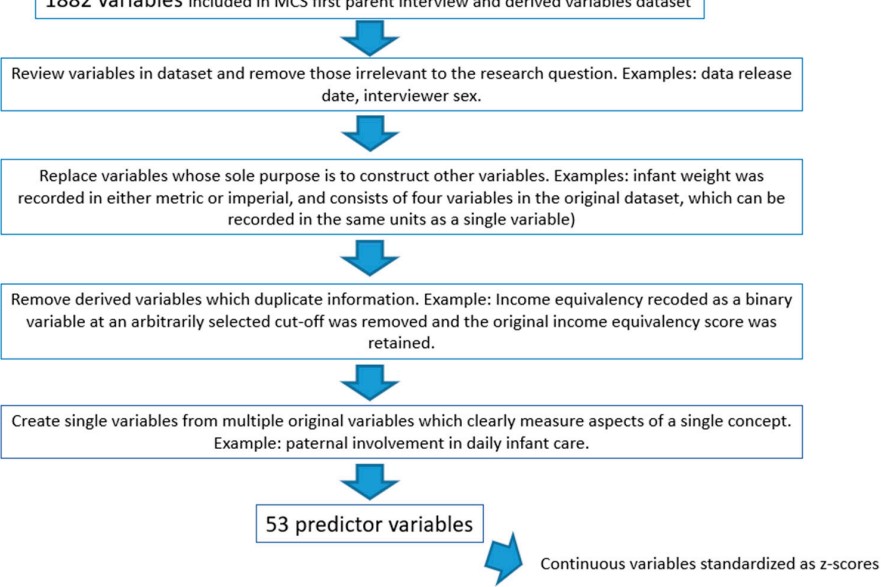

**Figure 1.** Variable selection procedure for RF algorithm.

*2.4. Data Analysis*

The MCS data were analysed using random forests (RF), a supervised machine learning decision tree algorithm easily implemented in statistical software such as Stata. In building each decision tree, the RF algorithm used half of the data (the training set) and with bootstrapping created sets of decision trees with the bootstrapped subsets of the data which comprise a decision rule at each branch node. The remaining half of the data for each tree (the test set) was used to test how well the algorithm performed classifying observations correctly. Missing data occurred due to unanswered interview items on a small number of variables, particularly paternal support. The RF algorithm contained a proximity algorithm to handle missing observations for features. Observations with a missing value for the outcome variable were dropped from the analysis, and continuous predictors were transformed to z-scores.

All analyses were carried out in Stata 16. For the RF model, the plugin Rforest was used [29]. Algorithm hypertuning of the number of variables included at each split and number of iterations were performed using Stata code developed by Schonlau and Zou [29]. Forwards elimination was applied to produce a reduced model which maximised the number correctly classified using the fewest variables.

## 3. Results

*3.1. Descriptive Statistics*

Descriptive statistics are displayed for all variables in Table 1, split into groups of variables as described above.

**Table 1.** Variable coding and descriptive statistics. All variables are from maternal or main care provider interviews.

| Variable | Coding | Obs | Mean (Std.Dev.) | Min–Max |
|---|---|---|---|---|
| Outcome and its constituent child development measures | | | | |
| Development below 2SD (age-adjusted) | | 18,432 | 0.038 (0.191) | 0–1 |
| Smiles | | 18,432 | 1.006 (0.082) | 1–3 |
| Sits up | | 18,432 | 1.066 (0.318) | 1–3 |
| Stands up holding on | | 18,432 | 1.475 (0.78) | 1–3 |
| Puts hands together | | 18,432 | 1.209 (0.532) | 1–3 |
| Grabs objects | | 18,432 | 1.01 (0.117) | 1–3 |
| Holds small objects | 1 = often, 2 = sometimes, 3 = not yet | 18,432 | 1.147 (0.454) | 1–3 |
| Passes a toy | | 18,432 | 1.065 (0.295) | 1–3 |
| Walks a few steps | | 18,432 | 2.81 (0.519) | 1–3 |
| Gives a toy | | 18,432 | 1.52 (0.717) | 1–3 |
| Waves bye-bye | | 18,432 | 1.912 (0.839) | 1–3 |
| Extends arms | | 18,432 | 1.205 (0.499) | 1–3 |
| Nods for yes | | 18,432 | 2.72 (0.617) | 1–3 |

**Table 1.** *Cont.*

| Variable | Coding | Obs | Mean (Std.Dev.) | Min–Max |
|---|---|---|---|---|
| Family and social support | | | | |
| Frequency mother sees her mother | 0 = lives with mother, 1 = every day, to 8 = never | 18,544 | 3.277 (2.352) | 0–8 |
| Mother has other parents to talk to | 1 = most, to 5 = least | 17,805 | 2.096 (1.016) | 1–5 |
| Family would help if financial problems | Strongly agree = 1 to strongly disagree = 5 | 17,803 | 1.747 (0.971) | 1–5 |
| Number of types of financial help from grandparents | Gifts, money for daycare, essentials, trust funds, household items, other | 18,547 | 1.235 (1.057) | 0–6 |
| Frequency mother reports spending time with friends | 1 = every day, to 5 = never or no friends | 18,527 | 2.958 (0.974) | 1–5 |
| Number of people who attended birth | | 18,432 | 1.12 (0.495) | 0–4 |
| Family-based infant care in work hours | 1 = no, 2 = yes | 18,387 | 1.17 (0.375) | 1–2 |
| Grandparent lives in household | 1 = yes, 2 = no | 18,432 | 1.921 (0.269) | 1–2 |
| Socioeconomic indicators | | | | |
| Equivalised household income | McClement's equivalised income | 18,432 | 296.833 (217.102) | 14.31–1250.78 |
| Age mother left full time education | | 18,341 | 17.578 (2.848) | 5–36 |
| Partner's SES from job | NS-SEC 7 classes, 1 = highest, 7 = lowest, 8 = not in work | 18,432 | 5.352 (2.641) | 1–8 |
| Partner's employment status | 1 = employed, 2 = self-employed, 3 = looking for work, 4 = not seeking work due to health, 5 = New Deal/apprenticeship, 6 = student, 7 = no partner/unknown | 18,432 | 3.388 (3.084) | 1–8 |
| Mother employed | Mother in paid work at 9 month interview = 1, else = 2 | 18,399 | 1.448 (0.497) | 1–2 |
| Winter temperature in room where baby sleeps | 5-point scale where 1 = warmest and 5 = cold | 18,310 | 2.301 (0.745) | 1–5 |
| Mother's report of pollution & grime in neighbourhood | Reported on a 4-point scale, 1 = most, to 4 = least pollution | 18,218 | 3.089 (0.892) | 1–4 |
| Infant characteristics | | | | |
| Infant's sex | 1 = male, 2 = female | 18,432 | 1.487 (0.5) | 1–2 |
| Infant has all immunisations | 1 = yes, 2 = no | 18,175 | 1.039 (0.194) | 1–2 |
| Infant's age in days when mother was interviewed | | 18,432 | 295.487 (15.23) | 243–382 |
| Infant's number of reported illness | | 18,422 | 1.633 (1.992) | 0–50 |
| Infant's number of accidents | | 18,430 | 0.083 (0.296) | 0–5 |
| Beliefs about parenting & parenting practices | | | | |
| Beliefs: Baby should be picked up when cries | 1 = strongly agree, to 5 = strongly disagree | 17,810 | 2.966 (1.045) | 1–5 |
| Beliefs: Stimulation is important for infant development | 1 = strongly agree, to 5 = strongly disagree | 17,806 | 1.431 (0.626) | 1–5 |
| Beliefs: Talking to infants is important | 1 = strongly agree, to 5 = strongly disagree | 17,814 | 1.200 (0.448) | 1–5 |
| Beliefs: cuddling infants is important | 1 = strongly agree, to 5 = strongly disagree | 17,815 | 1.191 (0.452) | 1–5 |
| Bed co-sleeping main sleeping arrangement in first 9 months | 1 = no, 2 = yes | 18,431 | 1.089 (0.285) | 1–2 |
| Breastfed at least 1 week | 1 = no, 2 = yes | 18,431 | 1.536 (0.499) | 1–2 |
| Work hours infant care is daycare centre | 1 = no, 2 = yes | 18,432 | 1.115 (0.319) | 1–2 |
| Main work hours infant care is mother | 1 = no, 2 = yes | 18,432 | 1.691 (0.462) | 1–2 |

**Table 1.** *Cont.*

| Variable | Coding | Obs | Mean (Std.Dev.) | Min–Max |
|---|---|---|---|---|
| | | Factors in pregnancy & birth | | |
| Birthweight (kg) | | 18,382 | 3.344 (0.589) | 0.39–7.23 |
| Estimated gestational age at birth (days) | | 18,201 | 275.727 (14.056) | 168–301 |
| Number of pharmacological pain interventions in labour | | 18,293 | 0.731 (0.667) | 0–4 |
| Infant conceived using fertility treatment | 1 = no, 2 = yes | 18,425 | 1.974 (0.159) | 1–2 |
| Duration of labour | In hours, C-section = 0 | 17,680 | 9.160 (11.145) | 0–100 |
| Type of delivery | 1 = normal, C-section & emergency = 2 | 18,398 | 1.313 (0.464) | 1–2 |
| Singleton birth | 1 = singleton, 2 = twin, 3 = triplet | 18,432 | 1.014 (0.123) | 1–3 |
| Pregnancy illnesses (e.g., preeclampsia) | 1 = yes, 2 = no | 18,396 | 1.623 (0.485) | 1–2 |
| Place of birth | Hospital = 1, else 2 | 18,401 | 1.020 (0.142) | 1–2 |
| How long mother and infant stayed in hospital after birth | 1 = weeks, 2 = days, 3 = hours | 18,020 | 2.046 (0.421) | 1–3 |
| Received full ante-natal care | 1 = yes, 2 = no | 18,391 | 1.038 (0.192) | 1–2 |
| | | Maternal factors | | |
| Mother's pre-pregnancy body mass index | | 16,813 | 23.649 (4.451) | 11.65–59.18 |
| Mother's birth year | | 18,426 | 1972 (5.95) | 1949–1987 |
| Mother reports being tired all the time | 1 = yes, 2 = no | 17,805 | 1.509 (0.5) | 1–2 |
| Mother reports being depressed | 1 = yes, 2 = no | 17,802 | 1.849 (0.358) | 1–2 |
| Average number of cigarettes mother smokes per day | | 18,420 | 3.315 (6.271) | 0–60 |
| Frequency mother drinks alcohol | Every day = 1 to never = 7 | 18,429 | 5.134 (1.49) | 1–7 |
| Mother has longstanding illness | 1 = yes, 2 = no | 18,425 | 1.789 (0.408) | 1–2 |
| Number of months pregnant at interview | | 18,423 | 0.196 (1.013) | 0–10 |
| | | Paternal & family factors | | |
| Ethnicity | 1 = white, 2 = mixed, 3 = India, 4 = Pakistani, 5 = Bangladeshi, 6 = Caribbean, 7 = African, 8 = East Asian & others | 18,402 | 1.627 (1.609) | 1–8 |
| Father present in household | 0 = yes, 1 = no | 18,403 | 0.172 (0.378) | 0–1 |
| Father's age when infant was born | | 18,395 | 31.91 (5.713) | 15–68 |
| Paternal involvement score: how much help father is | Summed score of how often father does: general childcare, feeding, getting up in night, changing nappies. 1 = least, to 21 = most | 16,255 | 10.205 (5.868) | 1–21 |
| Birth interval in months from older sibling | | 8997 | 42.803 (27.86) | 9–318 |
| Number of siblings in household | | 18,432 | 0.938 (1.081) | 0–9 |
| Mother reports partner sensitive and aware of her needs | Strongly agree = 1 to strongly disagree = 5 | 14,358 | 1.986 (0.929) | 1–5 |

### 3.2. RF Algorithms

After hyper-tuning to 23 variables at each split and 30 iterations, the RF algorithm for all 53 predictors had an out of bag error rate of 0.001. Only 19 of 18,432 infants were classified incorrectly, with all incorrect classifications being cases of delay which were not classified correctly (false negatives). Given that an algorithm with 53 features would take significant computing time to make a prediction for new cases, the algorithm was reduced by forwards selection beginning with the most important feature identified in the 53 feature algorithm, which was birthweight. Figure 2 displays the results of this process, in which including six features resulted in misclassification of 24 infants. Figure 3 displays feature importance scores in the 53-feature algorithm. Higher scores indicate higher importance in the algorithm. When running algorithms between 7 and all 53 features, there was only a very gradual drop in the number of cases misclassified (not shown in Figure 2), and thus, 6 features were deemed to successfully combine computational efficiency and classification accuracy. The top six features in relation with developmental delay are shown in Figure 4. Decision tree algorithms do not produce a statistic or parameter estimate showing the direction of association, as they are not linear models. To overcome this, two-way prediction plots are displayed for the reduced (six-variable) algorithm in Figure 4 and for all features in Supplementary Material Figure S1. The plots shown are two-way prediction plots with either a Lowess smooth fit line, a quadratic fit line or as a linear plot for binary predictors (whichever best described the observed relationship). The direction or shape of relationships between developmental delay and all predictors are described in writing in Figure 3.

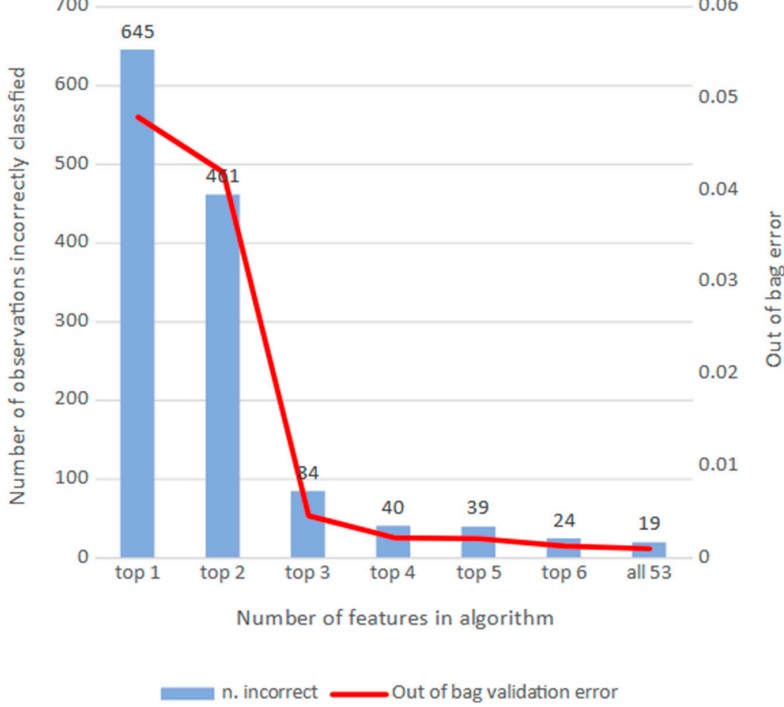

**Figure 2.** Number of correctly classified observations plotted with out of bag error for RF algorithms. The top 1 feature model includes birthweight only, and the top 2 includes birthweight and gestational age at birth. Beyond the six features with the highest importance scores in the full model, very little improvement in classification was evident.

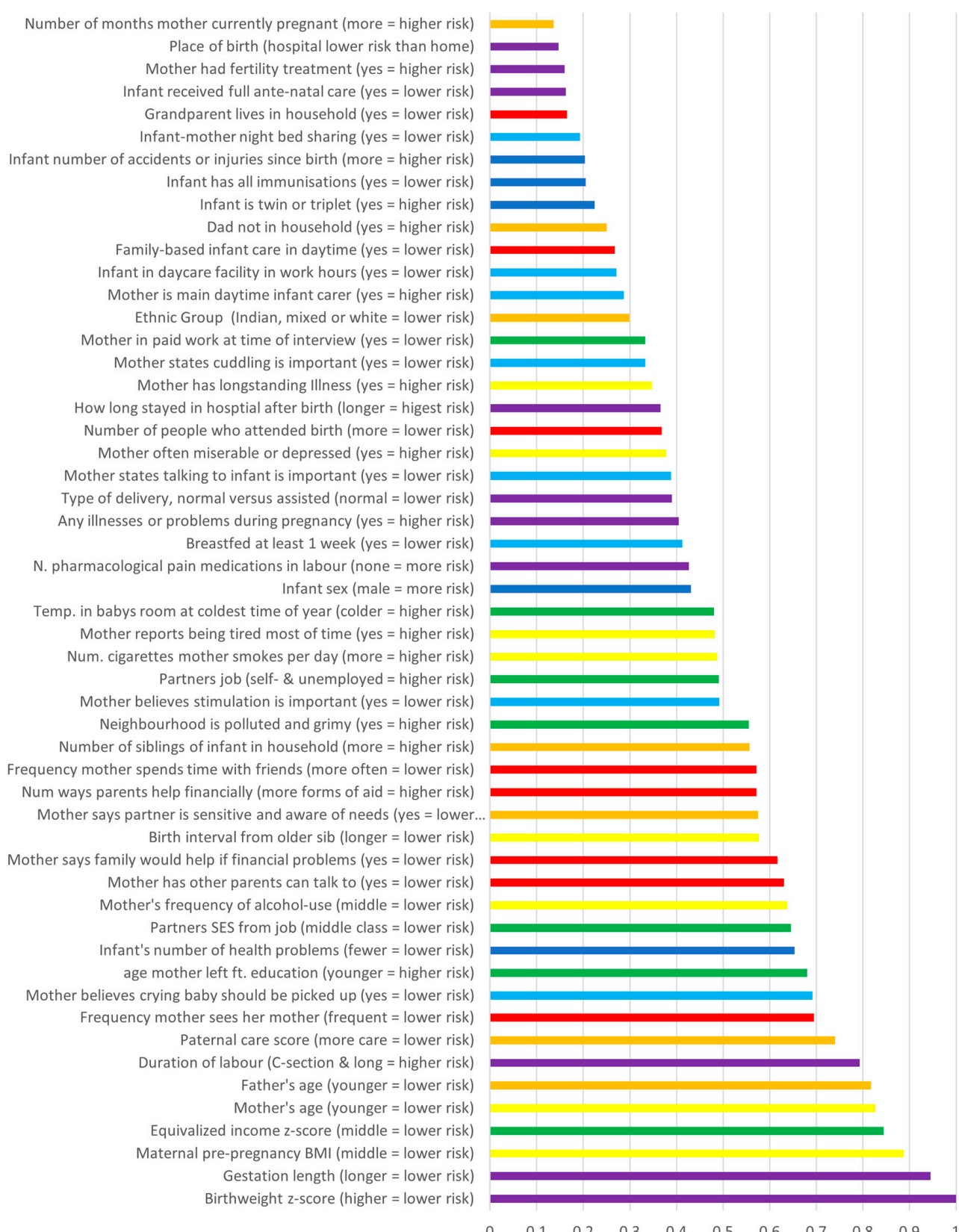

**Figure 3.** Importance plot using the feature importance scores from the 53-feature RF algorithm. Red bars = family and social support variables; green = socioeconomic indicators; dark blue = infant characteristics; light blue = beliefs about parenting; purple = medical factors in pregnancy and birth; yellow = maternal factors; orange = paternal and family factors.

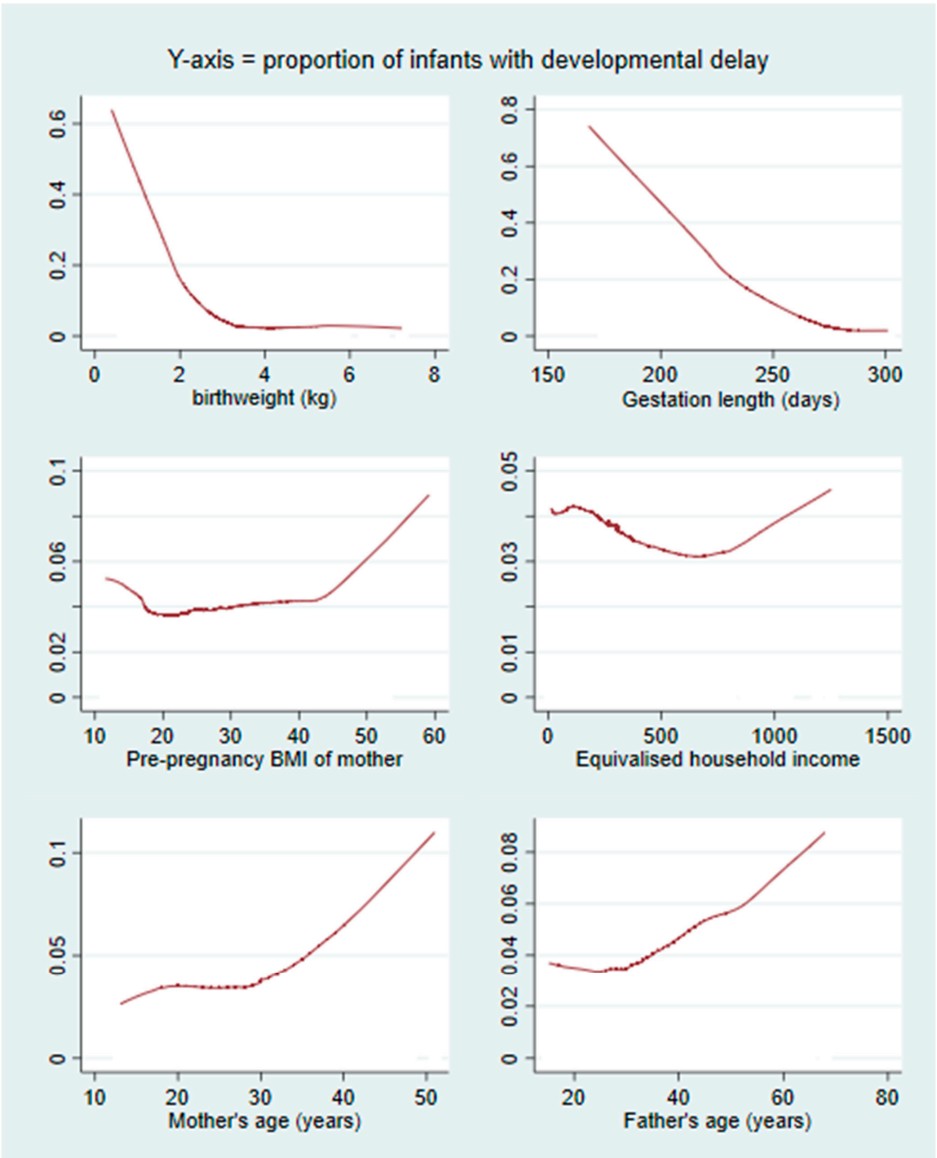

**Figure 4.** Two-way prediction plots displaying the shapes of the associations between the features with the highest importance scores and psychomotor delay. Lines are Lowess smoothed.

## 4. Discussion

### 4.1. Summary

The RF machine learning approach allowed simultaneous analysis of a large number of maternal, paternal, social and health-related factors. The algorithm performed very well when applied to the test data, with sensitivity at the level of a very good diagnostic medical test. The results were consistent with developmental delay having a complex aetiology: 47 variables had importance scores above 0.2. However, prediction did not improve substantially beyond the top six features, which were birthweight, the infant's gestational age at birth, maternal BMI, household income and maternal and paternal ages. None of the six most important features had linear relationships with developmental delay. Maternal pre-pregnancy BMI was a risk factor for developmental delay below a BMI of 18 and above 45. Household income had an important but complex nonlinear relationship with developmental delay.

*4.2. Comparisons to Prior Studies*

Preterm birth and low birth weight often occur together and are associated with at least a doubling of the odds of developmental delay [11–14] The results reported here showed that these two variables were not only the most important in the RF algorithms but were the most important variables by a substantial margin (see Figures 2–4). Figure 4 displays the proportion of infants with developmental delay at 9 months of age for the top six most important predictors. It shows that infants born below 1 kg had around a 50% probability of delay, while birth weights above 3 kg were associated with less than a 5% probability of delay. Preterm birth had a similarly large effect on the probability of delay (see Figure 4, top right).

While effect of low birth weight and preterm birth in the Millennium Cohort were consistent with patterns observed in prior research, the effects of socioeconomic indicators on developmental delay were not as clear. For example, Ozkan et al. [13] found that across the range of maternal education there was a tenfold increase in risk of delay. In the RF model, parental income was an important but non-linear predictor of delay which was associated with around a 3% risk of developmental delay at its lowest (middle income) and a 4% risk at low income (see Figure 4, centre right). This indicates that while income was a useful variable for classifying cases of delay in the Millennium Cohort despite having a non-linear association, the size of the effect was small. It should be noted that the increased risk of delay at high incomes visible in Figure 4 may be confounded if older mothers are more likely to have high household income.

Maternal age has previously been found to have the opposite relationship to developmental delay to what was found here: there was a monotonic trend towards lower risk of delay beginning with the youngest mothers (see Figure 4, bottom left). In prior research, infants of teenage mothers had an increased risk of delay [13]. Prior research additionally highlighted the importance of maternal education [16,17]. Here, income had a higher importance score than maternal education.

Pre-existing maternal obesity has been found to predict developmental delay in linear statistical models [15]. Maternal pre-pregnancy BMI was an important predictor of developmental delay using RF, but a nonlinear association was present in the Millennium Cohort, where increased risk is evident below a BMI of 18 and above 45. Risk was relatively constant between a BMI of 18 and 45, and the majority of women in the sample fell within this range: a BMI of 18 represents the third percentile, and a BMI of 45 represents the 99.8th percentile.

*4.3. Study Limitations*

A prospective longitudinal study design would be necessary to confirm algorithm performance in a clinical setting. Psychomotor delay in the MCS 9-month interview was measured using fewer items than are typically found in established scales such as Age and Stages. The same data quality issue applies more generally to most of the concepts in this analysis: national cohort study data allows for large analysis sample sizes and the potential for high statistical power, but this comes at a cost to the level of detail gathered about each concept: for example, family support variables were from interview rather than methods which directly measure social support. Methods that directly measure or change social support would be preferable.

**5. Conclusions**

RF can be easily implemented in statistical software such as Stata, as well as in open source software such as BlueSky Statistics and JASP. It is preferable to regression when there is a large number of potentially important predictors of an outcome and substantial nonlinearity in relationships between predictors and an outcome. A disadvantage is that other than producing classification accuracy, sensitivity and specificity values, the underlying concepts and results interpretation are not familiar to the majority of medical and social science researchers. The results of the RF modelling here showed remarkably

high sensitivity and specificity of almost 100%, which is far in excess of existing regression-based algorithms predicting developmental delay [15]. The features with the highest importance scores can all be discerned at birth: no features measuring aspects of early infant care or environment meaningfully added to algorithm performance. This implies that screening for developmental delay can be successfully implemented in the neonatal period. Maternal health problems during pregnancy, including eclampsia, bleeding and non-trivial infections also had lower importance scores than expected. This may be because eclampsia and other problems during pregnancy are associated with preterm birth and low birthweight rather than directly predicting developmental delay [30].

**Supplementary Materials:** The following supporting information can be downloaded at: https://www.mdpi.com/article/10.3390/reprodmed4020012/s1, Figure S1: Two-way prediction plots for all features; Table S1: Variable coding and data decsions.

**Funding:** This research received no external funding.

**Institutional Review Board Statement:** Detailed information on ethical approval can be accessed here: https://cls.ucl.ac.uk/wp-content/uploads/2017/07/MCS-Ethical-review-and-consent-Shepherd-P-November-2012.pdf (accessed on 1 November 2022).

**Informed Consent Statement:** Informed consent was given by the MCS cohort members. For details see: https://cls.ucl.ac.uk/wp-content/uploads/2017/07/MCS-Ethical-review-and-consent-Shepherd-P-November-2012.pdf (accessed on 1 November 2022).

**Data Availability Statement:** The data used in this study are available free of charge via the UK Data Service. https://beta.ukdataservice.ac.uk/datacatalogue/studies/#!?Search=&Rows=10&Sort=0&DataTypeFacet=Cohort%20and%20longitudinal%20studies&Page=1&DateFrom=440&DateTo=2022 (accessed on 15 January 2021).

**Conflicts of Interest:** The author declare no conflict of interest.

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
