# Peer review of "A Machine Learning Algorithm Predicting Infant Psychomotor Developmental Delay Using Medical and Social Determinants"

_2673-3897, doi:10.3390/reprodmed4020012_

Round 1

Reviewer 1 Report

1. A separate section to discuss the literature survey is required

2. What is the author's contribution in the work has to be highlighted at the end of introduction?

3. Only RF algorithm is used to develop the model. Comparison of results with other standard algorithms like, ANN, SVM, NB, k-NN etc., is required

4. Comparison with state-of-art approaches is essential.

5. The link provided to share the data is not working. 

Author Response

1. A separate section to discuss the literature survey is required.

Thank you, I have now separated the literature review in the introduction into sections.

2. What is the author's contribution in the work has to be highlighted at the end of introduction?

This is now included.

3. Only RF algorithm is used to develop the model. Comparison of results with other standard algorithms like, ANN, SVM, NB, k-NN etc., is required

Please see my response to question 4.

4. Comparison with state-of-art approaches is essential.

The intent was to use a machine learning approach that can be implemented in commonly-used statistical software rather than to compare many different AI approaches, which would require a much longer and more complex manuscript. Most university-based researchers would not consider applying AI in this area of the social and health sciences, but instead would rely on regression models in some form or other. My intent was to apply AI that is both approachable for researchers and powerful for prediction. I have added to the introduction to make these decisions clear. It should also be noted that the sensitivity and specificity values were so high that more sophisticated AI approaches such as neural networks could not add much if anything to the predictive power of the model. Their proneness to overfitting (fitting the model to statistical ‘noise’ or error in the data) is also not as well-established as it is for RF.

5. The link provided to share the data is not working. 

The link works for me. It should be noted that the data can’t be downloaded without first registering with the UK Data Service and agreeing to their terms of usage of data.

Reviewer 2 Report

A machine learning algorithm predicting infant psychomotor developmental delay using medical and social determinants. By David Waynforth

The paper shows the predictive power of perinatal variables, by including socio-economic predictors toward the child development. This sounds effective as a simplification is supposed to be offered from the statistical assessment. It is worth nothing that the AI seems the device to measure early the risk of the development and health of the child. The paper can be accepted.

Author Response

There were no changes suggested.

Round 2

Reviewer 1 Report

Literature review section is not yet separately discussed.

Author Response

I have added discussion as recommended (see lines 245-280).

Round 3

Reviewer 1 Report

nil